# Relevance of miR-223 as Potential Diagnostic and Prognostic Markers in Cancer [note 1]

**DOI:** 10.3390/biology11020249

**Published:** 2022-02-06

**Authors:** Faisal Aziz, Abhijit Chakraborty, Imran Khan, Josh Monts

**Affiliations:** The Hormel Institute, University of Minnesota, Austin, MN 55912, USA; aabhi.mn@gmail.com (A.C.); khan0672@umn.edu (I.K.); jmonts33@gmail.com (J.M.)

**Keywords:** biomarker, carcinogenesis, cell proliferation, circulating miRNA, diagnosis, miR-223

## Abstract

**Simple Summary:**

MicroRNAs (miRNAs) are endogenous small non-coding RNAs that function in the regulation of gene expression and regulate a wide array of biological processes, including carcinogenesis. Several mechanisms are involved in miRNA-associated cancer development, such as amplification or deletion of miRNA genes, abnormal transcriptional control of miRNAs, dysregulated epigenetic changes, and defects in miRNA biogenesis machinery. MiRNA-223 has been found to be a critical miRNA that is involved in a wide range of molecular processes. It is involved in the regulation of inflammatory cytokines, epithelial homeostasis, immune checkpoint signaling pathways, apoptosis, cell cycle, cell proliferation, invasion, and chemosensitivity. Published literature has demonstrated that miRNA-223 expression is associated with cancer development and prevention. Mir-223 functions as either a tumor suppressor or oncogene under certain circumstances, containing multiple targets or specific targets. Hence, miR-223 could be a potential candidate diagnostic biomarker, prognostic biomarker, or therapeutic target of cancer.

**Abstract:**

In 1993, the discovery of microRNAs in *Caenorhabditis elegans* (*C. elegans*) altered the paradigmatic view of RNA biology and post-transcriptional gene regulation. Further study revealed the role of microRNAs in disease development and progression. In particular, this review highlights microRNA-223 (miR-223 or miRNA-223) expression in malignant neoplastic disorders. miR-223 expression controls aspects of hematopoiesis and apoptosis, and cell proliferation, migration, and invasion. miR-223 regulates a number of gene targets, including cytoplasmic activation/proliferation-associated protein-1 (Caprin-1), insulin-like growth factor-1 receptor (IGF-1R), and other cell proliferation- and cell cycle-associated genes. Several studies have proposed miR-223 as a novel biomarker for early cancer diagnosis. Here, we emphasize miR-223′s role in the development and progression of cancer.

## 1. MicroRNAs (miRNAs)

MicroRNAs are short (~19–24 nucleotides in length), endogenous, non-coding RNAs that regulate protein expression by binding at complementary 3′-untranslated regions (UTRs) of targeted messenger RNA (mRNA) [1,2,3,4]. Binding within UTRs exerts post-transcriptional regulation of gene expression; these regulatory UTRs can affect polyadenylation, translation efficacy, and mRNA stability. miRNA association to mRNA interferes with translation by disrupting interactions between mRNAs and a number of translational factors. As a result, standard post-transcriptional gene expression is prevented or disrupted [5,6]. miRNA binding also triggers recruitment of mRNA decay factors, and their association results in decreased gene expression due to mRNA destabilization and degradation. In sum, microRNA downregulates gene expression of targeted mRNAs by inhibiting translation or by triggering transcript degradation [7,8,9].

miRNA biogenesis precedes the regulatory role of miRNA. First, primary microRNAs (pri-miRNAs) are transcribed by RNA polymerase II. Similar to most Pol II transcripts, they receive 5′-caps and 3′-polyadenlyated tails. In the nucleus, pri-miRNAs are processed into miRNA precursors (pre-miRNAs) by DiGeorge syndrome critical region gene 8 (DGCR8) and nuclear RNase III enzyme Drosha. This processing step generates the 5′-phosphate and 2-nucleotide 3′-overhang that are characteristic of pre-miRNAs. These miRNA precursors are transported into the cytoplasm by exportin-5. In the cytoplasm, Dicer1-TARBP2 complex produces mature miRNA duplex (18–25 nucleotides). The duplexed miRNAs are separated into guide and passenger strands based on their 5′-end base pairing. Typically, the miRNA strand with the least stable base pairing serves as the guide strand while the more stable passenger strand is usually degraded. Next, the guide strand, along with RNA-binding proteins, associates with Argonaute (AGO) proteins, forming a microribonuclear protein complex called RNA-induced silencing complex (RISC). The guide strand directs this complex to its mRNA target through sequence complementarity [7,10]. As mentioned previously, miRNA binding within complementary 3′-untranslated regions of mRNA leads to gene silencing via destabilization, degradation, and translational repression of mRNA [1,6,7,11].

## 2. General microRNA-223 Biology

As a hematopoietic-specific miRNA, microRNA-223 (miR-223 or miRNA-223) is essential for myeloid lineage development and function [6,12,13]. miR-223 acts during lineage commitment, promoting granulocytic differentiation and suppressing erythrocytic differentiation of progenitor cells [14]. Using transcriptome data, Fukao et al. proposed that miR-223 transcription is driven by myeloid transcription factors, PU.1 and C/EBPs [14]. miR-223 expression in myeloid lineage cells is well characterized in both humans and mice [15]. miRNA-223 is highly conserved in various organisms, underscoring its critical role in important physiological events [14]. miR-223 is associated with hematopoiesis, immune response, and different types of cancer development [16,17].

## 3. miR-223 and Cancer

miR-223 is typically repressed in hepatocellular carcinoma and leukemia [18,19], although higher expression levels of miR-223 are linked to colorectal [20] and recurrent ovarian cancers [21]. In some cases, miR-223 downregulation correlates with high tumor burden, disease aggressiveness, and poor patient prognosis. Consequently, understanding the complex role of miR-223 is essential for cancer diagnosis and treatment.

Several miR-223 targets are associated with malignancy: insulin-like growth factor-1 receptor (IGF-1R), monocytic enhancer factor 2C, microtubule destabilizer stathmin 1, and artemin and Forkhead box protein O1A (FOXO1A) [22]. For example, stathmin 1 plays a key role in cell cycle progression, chromosome segregation, and cell survival. Stathmin 1 overexpression has been observed in malignant hematopoietic cells, and stathmin 1 inhibition reportedly reduced the proliferation of leukemia cell lines. In sum, perturbation of stathmin 1 expression highlights the role of miR-223 in malignant neoplastic diseases [23]. According to Ma et al., sister chromatid cohesion protein PDS5 homolog B (PDS5B) is a direct target of miR-223 in prostate cancer. Furthermore, overexpression of PDS5B, which acts as a tumor suppressor gene, leads to the reversal of mR-223-mediated tumor progression in prostate cancer cells. Hence, the miR-223/PDS5B complex plays a critical role in the regulation of cell proliferation and invasion in prostate cancer cells [24]. In general, various hallmarks of cancer are induced by miRNA-223, including cell proliferation, tumorigenesis, and metastasis. In contrast, anti-miR-223 attenuates the invasiveness and proliferation of some cancers. Anti-miR-223 also stimulates apoptosis.

Consequently, miR-223 plays a significant role in cancers, such as gastric, hepatocellular, ovarian, lung, and esophageal cancers [25,26]. miR-223 downregulation is correlated with the development of chronic lymphocytic leukemia and primary small cell lung cancer (SCLC) [26,27] while its upregulation is associated with ovarian and colorectal cancers. In particular, miR-223 is regarded as a possible biomarker in recurrent ovarian cancer. For future diagnostic strategies, miR-223 expression can be used to distinguish carcinomas from noncancerous lesions. microRNA-223 can also be considered a therapeutic target for cancer treatment [27] (Figure 1).

### 3.1. Gastric Cancer

Gastric cancer is a multifactorial disease with a complex interplay between genetics, lifestyle, and environment [28,29,30,31]. *Helicobacter pylori* infection is the most common chronic bacterial infection and affects over 50% of the world’s population. It is a major cause of gastritis and gastric ulcers, and gastric cancer [32,33,34]. Recent gene profiling studies identified a link between miRNAs and gastric-related disease. Compared to healthy individuals, miRNA expression was significantly different in patients with chronic gastritis and gastric cancer [2]. Gastric cell lines with elevated miR-223 levels displayed a significant reduction in apoptosis. miR-223 overexpression also activated gastric cancer cell proliferation and invasion. Overall, miR-223 may play a crucial role in tumor progression and the survival of gastric cancer cells. As a result, miR-223 is proposed as an oncogene for gastric cancer [27]. As such, miR-223 expression can assist in the diagnosis and treatment of gastric cancer [27].

The mechanism responsible for increased gastric carcinogenesis was reported by Li et al., 2011. Their work demonstrated that miR-223 binds to FBXW7/hCdc4 at the 3′-UTR region of FBXW7/hCdc4 mRNA. The FBXW7/hCdc4 gene is a well-established p53-dependent tumor suppressor gene that encodes an ubiquitin ligase involved in chromosome stability. FBXW7 is a commonly deregulated ubiquitin-proteasome system protein in human cancers. As follows, miRNA-223 downregulation of FBXW7/hCdc4 is involved in gastric tumorigenesis due to increased genetic instability. In summary, miR-223 exerts its carcinogenic role by downregulating FBXW7/hCdc4 expression. This outcome rationalizes micrRNA-223 as a novel therapeutic target for gastric cancer [2,27].

### 3.2. Hepatocellular Carcinoma

Hepatocellular carcinoma (HCC) is the third leading cause of cancer mortality worldwide, and the number of cases and deaths is predicted to increase [35]. HCC patients showed significantly higher miR-223 levels compared to healthy individuals. As a result, miR-223 is proposed as a potential diagnostic biomarker for HCC. However, tissue injury, more than carcinogenesis, is implicated in miR-223 overexpression for hepatocellular disorders. Hepatitis B patients with HCC-negative results showed higher serum levels of miR-223 compared to HCC and healthy subjects. Chronic hepatitis patients displayed greater hepatocyte damage compared to HCC patients, and hepatitis (tissue injury) likely leads to elevated miRNA-223 serum levels rather than HCC [36].

### 3.3. Breast Cancer

Breast cancer is a major cause of mortality among women in developed and developing countries (IARC, 2008) [37,38]. Its incidence is continuing to increase, likely due to early screening and its public health emphasis, along with other factors [39,40]. According to Gong et al., breast cancer cells (MDA-MB-231) express abnormally low levels of miR-223 and abnormally high levels of Caprin-1. Caprin-1 overexpression promotes proliferation and invasion of breast cancer cells. Conversely, miR-223 expression inhibits breast cancer cell proliferation and invasion in MDA-MB-231 cell lines [41]. miR-223 targets 3′-UTR of Caprin-1 mRNA and downregulates Caprin-1 expression. A decrease in miR-223 biogenesis likely reduces Caprin-1 regulation. Caprin-1 deregulation then results in greater carcinogenesis in breast cancer cell lines, such as MDA-MB-231 [41]. In summary, miR-223 may suppress proliferation and invasion of breast cancer cells by directly targeting Caprin-1. As a result, miR-223 and Caprin-1 expression can be used predictively in cancer diagnosis and prognosis [41].

### 3.4. Lung Cancer

Lung cancer is an aggressive disease, particularly when considering its high incidence and mortality rates [42]. Consequently, novel therapeutic biomarkers are needed to improve disease outcomes [43]. Disease progression displays a negative correlation between miR-223 expression and lung cancer development and metastasis [44,45,46]. According to Nian et al., miR-223 functions as a potent tumor suppressor by targeting IGF-1 receptor and cyclin-dependent kinase 2. Low miR-223 serum levels are associated with high cancer-related death. As a result, miR-223 can be used as an early diagnostic and therapeutic biomarker to improve the treatment of lung cancer [22].

### 3.5. Ovarian Cancer

Ovarian cancer is a heterogeneous disease that encompasses a number of different cellular subtypes. The most common subtype is high-grade serous ovarian cancer (HGSOC). HGSOC is also the most lethal of all gynecologic malignancies diagnosed in the United States [47]. Recent advances have been unable to significantly improve patient outcomes. In the last 20 years, patient survival at 5 years post-diagnosis has marginally improved to 46% from about 35–38% [48].

miR-223-3p expression is upregulated in ovarian cancer. In this context, miR-223 decreases sex-determining region Y-box 11 (SOX11) expression. Inhibiting the *sox11* gene negatively regulates cell proliferation, migration, and invasion. Accordingly, a therapeutic target of miR-223-3p can potentially regulate ovarian cancer by targeting SOX11 expression [48].

### 3.6. Osteosarcoma

Osteosarcoma is a bone tumor malignancy in adolescents and young adults [49]. miR-223 is downregulated in osteosarcoma [50,51]. Xu et al. reported that upregulation and downregulation of miRNA-223 lead to significantly diminished and enhanced cell proliferation and cell cycle progression in osteosarcoma, respectively. Thus, miR-223 is proposed as a therapeutic biomarker for osteosarcoma [51].

### 3.7. Other Cancers

miRNA dysregulation is frequently observed in colon cancer [52]. Previous studies found that the miR-223 oncogene is upregulated in colon cancer [53,54]. In 2017, Liu et al. proposed that miR-223 promotes colon cancer invasion and metastasis by downregulating p120, which reduces intercellular adhesion, promotes RhoA activity, and activates β-catenin signaling [53,55]. As a result, miR-223 is a potential diagnostic and therapeutic target for anti-colon cancer treatment. It was reported that miR-223 functions as an oncomiR in T cell acute lymphoblastic leukemia, whereas in acute myeloid leukemia (AML), it functions as a tumor suppressor [56,57,58]. According to Gao et al., low expression of miR-223 has been found in various human hematologic malignancies and solid tumors, where it acts as a tumor suppressor, and has a significant role in leukemia and lymphomas. miR-223 significantly inhibits the proliferation, growth rate, and colony formation of cells in vitro, and tumor formation in vivo through targeting IGF-1R and its downstream PI3K/Akt/mTOR/p70S6K pathway in leukemia [57,59,60]. Compared with normal tissues, miR-223 is also upregulated within the cancerous tissues of pancreatic, gastric, and prostate cancer biopsies [55,61]. In a study by Wei et al., miR-223 promoted prostate tumor development and malignancy by decreasing cell apoptosis and increasing cell migration. These effects were reversed by upregulating SEPT6. As a gene target of miR-223, SEPT6 is downregulated by miR-223 overexpression. Consequently, SEPT6 can serve as a potential therapeutic target for prostate cancer [62].

## 4. miR-223 and Key Motifs of Malignancy 

Many studies have reported abnormal microRNA-223 expression in different types of cancers. These studies highlight the importance of miR-223 in key motifs of malignancy: aberrant cell proliferation, invasion, and metastasis [63,64].

### 4.1. Proliferation and Cell Cycle Progression Are Affected by miR-223

Several studies have investigated the role and mechanism of miRNAs in cancer cell proliferation and apoptosis [65,66]. miRNAs typically regulate genes that are involved in cell growth, proliferation, and apoptosis rates. According to Guz et al., genes responsible for cell cycle progression and differentiation are often downregulated in cancer cells while other regulatory genes involved in cell cycle progression and resistance to apoptosis are overexpressed [67]. According to Xu et al., overexpression of miR-223 leads to reduced FBXW7 mRNA levels and increases endogenous cyclin E protein, which in turn increases genomic instability. In contrast, a low level of miR-223 level leads to increased Fbw7 expression and decreased cyclin E activity. Furthermore, it was proved that miR-223 expression is reactive to acute alterations in cyclin E regulation by the Fbe7 pathway [68]. Recent evidence demonstrated that aberrant miRNA expression is critical in this malignant phenotype [69,70]. In particular, miR-223 was a key feature in cancer cell proliferation [69,70,71].

1.Effect of miR-223 on FOXO1 and cell proliferation

Forkhead box O (FOXO) transcription factors FOXO1, FOXO3a, FOXO4, and FOXO6 are proteins that regulate genes involved in apoptosis, DNA damage repair, cell cycle progression, and more. Ultimately, these FOXOs promote or repress multiple gene targets that are involved in tumor suppression. These targets include Bim, Trail, and Fas L, which induce apoptosis; p21, p27, and cyclin D1, which regulate cell cycle progression; and GADD45a, which supports DNA damage repair [64].

According to Wu et al., miR-223 regulates FOXO1 and its phosphorylation state [64]. Low miR-223 levels are observed in HuH7 and HCT116 colorectal cancer cell lines. Upon miR-223 overexpression, cytosolic levels of FOXO1 mRNA and protein were significantly downregulated. More specifically, the inactive form of phosphorylated FOXO1 was downregulated after miR-223 overexpression, and the active unphosphorylated form of FOXO1 was upregulated. As unphosphorylated active FOXO1 accumulated, the protein became increasingly localized in the nucleus. Nuclear FOXO1 activation increased p21 expression and cell cycle arrest [64]. Wu et al. demonstrated that miR-223 regulates FOXO1 and its downstream targets: p21, p27, and cyclin D1. Increased p21 and p27 expression, or decreased cyclin D1 expression, consistently coincided with cell growth retardation upon miR-223 overexpression. In HuH7 and HCT116 cell lines, proliferation was significantly slowed as well. As a result, Wu et al. established a role for miR-223 in the inhibition of cancer cell proliferation via FOXO1 regulation [64].

2.Regulation of IGF-1R by miR-223 to modulate cell proliferation

Insulin-like growth factor-1 receptor (IGF-1R) and its ligands regulate cell proliferation and apoptosis. IGF-1R deregulation has been widely documented in the progression of human malignancies [72,73]. According to Jia et al., miR-223 expression inhibits hepatoma cell proliferation, growth rate, and colony formation by targeting IGF-1R for downregulation. The downstream pathway, Akt/mTOR/p70S6K, is inhibited as well. In sum, IGF-1R is a functional target for miR-223-directed suppression of cell proliferation [74].

3.Role of miR-223 on E2F1 and cell proliferation

The E2F family of transcription factors regulate cell cycle progression by transactivating cell proliferation genes [75]. In particular, E2F1 overexpression leads to enhanced proliferation of cancer cells [75,76]. According to Pulikkan et al., miR-223 and E2F1 transcription factor act within an autoregulatory negative feedback loop in acute myeloid leukemia (AML). In brief, E2F1 induces C/EBPα expression, which subsequently induces miR-223. In turn, miR-223 represses E2F1 expression. However, AML exhibits low miR-223 expression, and E2F1 is responsible for this phenotype. E2F1 binds to the miR-223 promoter in AML blast cells. This association is responsible for transcriptional repression of the miR-223 gene and the low miR-223 expression observed in AML [75,76]. Upregulation of miR-223 would reverse the effects of E2F1 overexpression and suppress cell proliferation in AML [76].

4.Effect of miR-223 in cytoplasmic activation/proliferation-associated protein-1 (Caprin-1)-induced cell proliferation

Cytoplasmic activation/proliferation-associated protein-1 (Caprin-1) is associated with cell proliferation. Caprin-1 downregulation leads to reduced cell proliferation and a prolonged cell cycle phase (G1 phase) [77,78]. According to Gong et al., miR-223 interacts with Caprin-1, and this interplay may be associated with the phenotypic proliferation and migration patterns observed in breast cancer cells with low miR-223 levels. Overall, miR-223 overexpression leads to decreased Caprin-1 expression, which in turn represses the proliferation and invasion of breast cancer cells [41].

### 4.2. Metastasis and Invasion of Cancer

Several studies reported the role of miR-223 in tumor migration and invasion, and miR-223 is considered a metastasis-promoting miRNA. For example, miR-223 regulates certain transcription factors to promote gastric cancer cell migration and invasion in vivo and in vitro. According to Li et al., miR-223 overexpression facilitates gastric cancer progression to invasive and metastatic stages. In contrast, its silencing leads to inhibited gastric cancer cell migration and invasion [2]. Haneklaus et al. also concluded that miR-223 overexpression may promote a shift to metastasis [5].

Li et al. investigated the regulatory role of miR-223 in esophageal cancer cell migration and invasion. They found that esophageal cancer cells display low miR-223 and high artemin levels. Subsequent miR-223 overexpression inhibited artemin levels and suppressed cell migration and invasion [79].

## 5. miR-223 as a Circulating Noninvasive Biomarker

miRNAs can serve as effective biomarkers for cancer diagnosis [36,80]. Tissue miRNA expression profiles can be used as diagnostic and therapeutic biomarkers for cancer [81]; however, their relative inaccessibility and invasiveness make tissue miRNA levels less suitable for diagnostic testing compared to freely circulating miRNA in plasma or serum. The greatest advantage of circulating miRNAs is their capacity to be non-invasively sampled via blood collection. Circulating miRNA detection could also provide greater sensitivity and specificity compared to current serum biomarkers, such as CA19-9 and CEA [82,83]. Consequently, circulating miRNAs are promising early diagnostic tools. According to Chang et al., miR-233 found in both clinical samples in plasma and stool significantly diagnoses colorectal cancer patients. Furthermore, stool and plasma clinical samples yield the highest sensitivity (96.8%) and specificity (75%), with an accuracy of 0.907 [84]. According to Li et al., miR-223 levels are significantly higher in gastric cancer plasma samples. As a result, microRNA-223 can serve as a biomarker for gastric cancer. Several other types of cancer show high miR-223 expression as well, including esophageal and hepatocellular carcinomas. In summary, circulating miR-223 can potentially serve as a novel noninvasive biomarker in cancer diagnosis [80] (Table 1).

## 6. miR-223 Application as a Diagnostic and Therapeutic Biomarker for Cancers

Cancer treatment is challenging for many reasons, including clinical and tumor heterogeneity and the lack of disease-specific diagnostic biomarkers [85,86,87,88]. Certain microRNAs have tumor suppressor or pro-oncogenic functions, making them attractive targets for cancer therapy [74]. Significant advances have been made in miRNA-based therapy; however, many challenges must be overcome before clinical usage [89]. Regardless, several studies reported miR-223 expression as a potential diagnostic and therapeutic target [36,89,90,91].

### 6.1. Hepatocellular Carcinoma and Liver Damage

Hepatocellular carcinoma (HCC) displays a high recurrence rate. This common reoccurrence requires further investigation into diagnostic and therapeutic targets for HCC treatment [92]. miR-223 overexpression was reported in patients with either HCC or chronic hepatitis B. The usefulness of miR-223 as a diagnostic biomarker for HCC is limited by its shared miR-223 profile. Chronic hepatitis B also displays elevated miR-223 levels; however, this overexpression is caused by inflammation and liver injury, not the presence of hepatocellular carcinoma [36,93,94].

According to Karakatsanis et al., miR-223 downregulation inhibits HCC cell growth, suggesting that miR-223 plays an important role in HCC progression and metastasis [94] Moreover, it was reported that miR-223-3p could be used as a promising prognostic biomarker as circulating miR-223-3p significantly differentiates HCC from the non-HCC groups [95]. Consequently, further investigation of miR-223 is essential.

### 6.2. Breast Cancer

Escobar et al. highlighted the effects of macrophages and miRNA on breast cancer cells. miR-223 overexpression, coupled with high levels of macrophage infiltration, results in an aggressive cancer phenotype [91]. In 2018, Yang et al. showed that miR-223 inhibits the proliferation and invasion of breast cancer by targeting STIM1 [96]. According to Chen et al., high expression of miR-223 acts as a tumor suppressor, which is correlated with prolonged survival in triple-negative breast cancer patients. Hence, miR-223 could be an independent prognostic biomarker in breast cancer [97]. As reported, miR-223 can serve as a potential diagnostic biomarker for breast cancer [91,96].

### 6.3. Esophageal Cancer

Esophageal cancer is a leading cause of cancer-related deaths worldwide. Most physicians recommend surgical resection for esophageal carcinomas at their early stages [98]. Due to the limited treatment options, further research is needed to identify biomolecular pathways involved in esophageal cancer malignancy [90,98,99].

Artemin (ARTN) is a growth factor associated with the tumorigenesis and progression of human cancers. ARTN also promotes cell migration, invasion, and metastasis [99,100,101]. After chemotherapy, artemin expression is strongly correlated with relapse and death in carcinoma patients [102,103].

According to Wu et al., miR-223 directly binds the ARTN 3′-UTR and regulates ARTN protein expression [102]. ARTN function is suppressed by miR-223 overexpression, which reduces esophageal cancer cell migration and invasion. In contrast, ARTN upregulation via miR-223 silencing promotes cell migration and invasion [102]. These observations rationalize microRNA-223 and artemin as therapeutic targets for esophageal cancer treatment.

### 6.4. Lung Cancer

In lung cancer, like all cancers, early diagnosis promotes favorable patient outcomes [103]. As a result, the identification of novel early diagnostic biomarkers related to tumorigenesis is vital for future advancements in cancer diagnosis and treatment [43].

miR-223 targets IGF-1R, a transmembrane receptor tyrosine kinase related to lung cancer oncogenesis, tumor growth, and cancer cell survival [103]. IGF-1R upregulation is involved in the initial dysregulation that shifts previously healthy cells towards tumorigenesis and malignancy. In contrast, IGF-1R downregulation sensitizes lung cancer cells to chemotherapy and radiation by inhibiting cell proliferation. As reported, miR-223 overexpression leads to tumor suppression in lung cancer [103]. According to Antona et al., miR-223 could be an early-stage serum biomarker for non-small cell lung cancer (NSCLC). Moreover, miR-223 is considered as an effective and reproducible serum biomarker [104]. Overall, miR-223 targets IGF-1R and its downstream signaling pathway for downregulation, leading to inhibited cell growth and proliferation [74]. Targeting miR-223 for overexpression could be a good strategy for lung cancer treatment.

## 7. Concluding Remarks

This review highlights the importance of miR-223 in cancer biology. miR-223 expression potentially plays a major role in cancer diagnosis, progression, and treatment. Data reported within this review characterizes miR-223 as emerging triple-negative breast, gastric, lung, and ovarian cancer biomarkers. Several genes and gene products were identified as targets of miR-223, including NLRP3, IL18, IL1β, DNA methyltransferase 1, and checkpoint kinase-1, signifying its potential use as a cancer-specific diagnostic biomarker and therapy. However, further investigation is needed to better understand the mechanisms of miR-223 involved in tumor activation or suppression. Nonetheless, miR-223 regulation is a promising tool for controlling key cancer motifs: increased cell growth, proliferation, and invasiveness in addition to decreased apoptosis rates.

## Figures and Tables

**Figure 1 biology-11-00249-f001:**
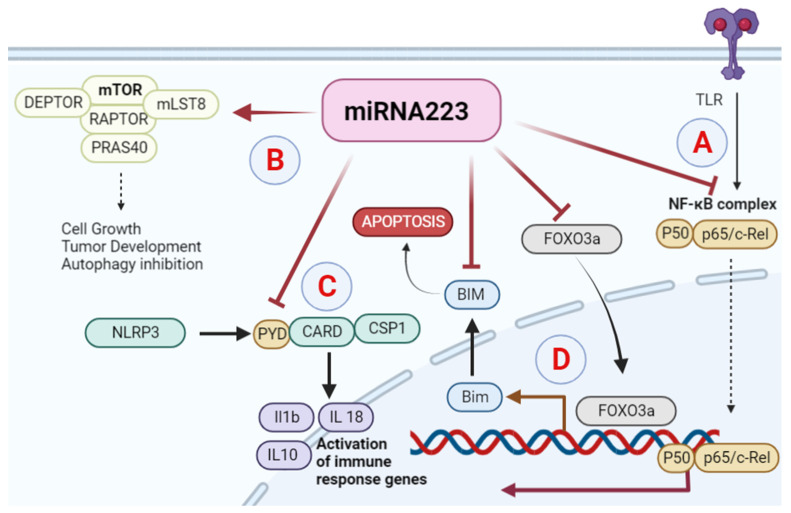
Role of miRNA223 in different cancers: (**A**) activated miR-223 negatively regulates the TLR4/MyD88-NF-κB signaling pathway. (**B)** miR-223 plays a role as a tumor suppressor in hepatocellular carcinoma by apoptosis through mTOR pathway activation. (**C**) miR-223 inhibits NLRP3 and its downstream factor CARD-PYD in breast cancer. (**D**) In colon cancer, miR-223 downregulates the transcription factor FoxO3a and BAM.

**Table 1 biology-11-00249-t001:** The roles of MiR223 in the regulation of carcinogenesis by targeting specific targets.

Clinical Relevance	Molecular Mechanism Involved or Function	Experimental Model	Specific Targets	References
Breast cancer	miR-233-3p mimics inhibited the NLRP3-dependent processes in cancer cells by suppressing the NLRP3 and downstream factors, including PYD and CARD domain-containing protein, IL-1β, and IL-18.	**Breast cell lines**: HMEC, MDA-MB231, MCF-7, and SKBR3 **Xenograft mouse model**	NLRP3, PYD, CARD, IL1β, IL18	PMID: 30747211
Hepatocellular carcinoma (HCC)	Checkpoint kinase 1, DNA methyltransferase 1, baculoviral IAP repeat-containing 5, kinesin family member 23, and collagen, type I, α1 target genes were considered the hub genes of miR-223-3p in HCC, which are significantly upregulated in HCC	**Database**: GEO database, TCGA database, Meta-analysis., Bioinformatics evaluation, Protein–protein interaction (PPI) network construction	Checkpoint kinase 1, DNA methyltransferase 1, baculoviral IAP repeat-containing 5, kinesin family member 23, and collagen, type I, α1.	PMID: 29207133
Prostate Cancer	miR-223 inhibited the malignant behavior of prostate cancer cells while EYA3/c-Myc had the opposite effect. Moreover, circGNG4 enhanced the expression of EYA3/c-Myc by sponging miR-223 to promote the growth of prostate cancer tumors in vivo.	**Prostate clinical samples**: Prostate Cancer and adjacent normal tissues. **Prostate cell lines**: PC-3, LNCaP, VCaP, and DUL145, and the human normal prostatic epithelial cell line, RWPE-1 **Xenograft mouse model**	circGNG4, EYA transcriptional coactivator, phosphatase 3 (EYA3)/c-Myc	PMID: 34395419
Esophageal squamous cell carcinoma (ESCC)	High expression level of miR-223 had a significant adverse impact on the survival of ESCC patients through repression of the function of FBXW7.	**ESCC clinical samples**: ESCC Primary ESCC tissue samples and human **ESCC cell lines**: TE1, TE4, TE6, TE8, TE9, TE10, TE14, and TE15.	FBXW7	PMID: 22108521
Gastric cancer	miR-223 induced by the transcription factor Twist, post-transcriptionally downregulates EPB41L3 expression by directly targeting its 3′-untranslated regions.	**Gastric clinical samples**: Gastric cancer and normal gastric tissues, **Gastric cell lines**: HEK293 cell, GPG29 cell, immortalized normal gastric mucosa GES cell, normal stomach fibroblastic cell NSFC, non-metastatic gastric cancer SUN-1, KATO-III, NUGC-3 cells, and metastatic gastric cancer XGC-9811L, AGS, and N87.	EPB41L3	PMID: 21628394
Hepatocellular carcinoma (HCC)	miR-223 functions as a tumor suppressor and plays a critical role in inhibiting the tumorigenesis and promoting the apoptosis of HCC through the mTOR signaling pathway by targeting Rab1.	**Hepatocellular cell lines**: Huh7, Hep3B, Bel-7402, and the HepG2 cell lines	Rab1, mTOR	PMID: 27998765
Lung cancer	miR-223 targets IGF-1R, a transmembrane receptor tyrosine kinase related to lung cancer oncogenesis, tumor growth, and cancer cell survival. miR-223 overexpression leads to tumor suppression in lung cancer		IGF-1R,	PMID: 10857553
Liver cancer	miR-223-3p down-regulated the expression of FAT1, and inhibited the proliferation, migration, invasion, and EMT of liver cancer cells by targeting FAT1. FAT1 was highly expressed in liver cancer tissues and cells while miR-223-3p was lowly expressed.	**Liver clinical samples**: Liver Tumor tissues and corresponding adjacent tissues. **Liver cell lines**: Normal human hepatocyte, L-02 and human liver cancer cell lines, Hep G2, SMMC-7721, Hep 3b, and Li-7,	FAT1	PMID: 32233593
Brain cancer	miR-223-3p could be decreased by NLRP3 overexpression, which was considered as one of target genes of miR-223-3p. miR-223-3p might act as a suppressor and a potential therapy target of glioblastoma.		NLRP3	PMID: 30033329
Colon cancer	miR-223 regulate the FoxO3a/BIM signaling pathway and colorectal cancer cell proliferation and apoptosis. Downregulating the expression of miR-223 increased the expression of FoxO3a and BIM and weakened cell proliferation and induced apoptosis.	**Colon cell lines**: NCM460 and SW620 cancer cells.	FoxO3	PMID: 29949152

## Data Availability

Not applicable.

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
