# Peer review of "Relevance of miR-223 as Potential Diagnostic and Prognostic Markers in Cancer [Author-notes fn1-biology-11-00249]"

_biology, 2022, doi:10.3390/biology11020249_

Round 1
Reviewer 1 Report
Presented review dealing with the discussion of the role of miR-223 in different aspects (cell proliferation, migration, and invasion) in relation to the different types of cancer and leukemia. Authors describe how miR-223 targets modulate cell proliferation and cell cycle in cancer. The review mentions also the important role of miR-223 as a diagnostic and therapeutic biomarker in different cancers/leukemia.
The main contribution of this review is the description of the complex role of miR-223 in development and progression of cancers and leukaemia’s.
General comments:
Review is logically structured. The topic of the review is relevant and important for the field of miRNAs as cancer/leukaemia biomarkers.
However, there are many reviews with a similar topic (the miR-223 in cancer) today. To make the present review noticeable and more effective for readers, I recommend strongly to summarize reviewed knowledge by using tables that will describe role of miR-223 in different cancers/leukaemias (especially, with miR-223 targets and their relation to cancers/leukaemias). Good will be to provide pictures about miR-223 implication in reviewed biological pathways Authors are welcomed to find inspiration in the recent review with the similar topic - Peng Jiao et al, miR-223: An Effective Regulator of Immune Cell Differentiation and Inflammation, Int. J. Biol. Sci. 2021, Vol. 17.
Specific comments:
1) Authors in their review must correct the citations, as there are several mistakes:
- Line 97: missing the year of reference Li et al…
- In the references list, the references with numbers 8, 19, 50, 59 and 62 have incorrect first author name - e.g. ref 8 Biology, C.; ref 19 Knowledge, B.C.; Practices, S.; …and so on. Please, correct through the list (and double check in the manuscript text).
2) Concluding remarks seems to be too general. I recommend specifying in more details e.g in line 313 …in which cancers the miR-223 is emerging biomarker.
Author Response
Reviewer 1.
Comments and Suggestions for Authors
Presented review dealing with the discussion of the role of miR-223 in different aspects (cell proliferation, migration, and invasion) in relation to the different types of cancer and leukemia. Authors describe how miR-223 targets modulate cell proliferation and cell cycle in cancer. The review mentions also the important role of miR-223 as a diagnostic and therapeutic biomarker in different cancers/leukemia. The main contribution of this review is the description of the complex role of miR-223 in development and progression of cancers and leukemia’s.
Specifics:
Comment 1. I recommend strongly to summarize reviewed knowledge by using tables that will describe role of miR-223 in different cancers/leukaemias (especially, with miR-223 targets and their relation to cancers/leukaemias).
Response 1. Thank you for your valuable advice. We have made a table summarizing the reviewed knowledge.
Comment 2. Good will be to provide pictures about miR-223 implication in reviewed biological pathways Authors are welcomed to find inspiration in the recent review with the similar topic - Peng Jiao et al, miR-223: An Effective Regulator of Immune Cell Differentiation and Inflammation, Int. J. Biol. Sci. 2021, Vol. 17.
Response 2. Thank you for your kind suggestions. We have drawn a figure explaining biological pathways by getting inspiration from the suggested recent review.
Comment 3. Authors in their review must correct the citations, as there are several mistakes: Line 97: missing the year of reference Li et al…In the references list, the references with numbers 8, 19, 50, 59 and 62 have incorrect first author name - e.g. ref 8 Biology, C.; ref 19 Knowledge, B.C.; Practices, S.; …and so on. Please, correct through the list (and double check in the manuscript text).
Response 3. Thank you for your corrections. We have corrected the references as, line 97, and references number 8, 19, 50, 59 and 62. Furthermore, we have double checked all the references and corrected it.
Comment 4. Concluding remarks seems to be too general. I recommend specifying in more details e.g in line 313 …in which cancers the miR-223 is emerging biomarker.
Response 4. Thank you for your valuable advice. We have modified the conclusion by mentioning the specific cancers type as well as their significance in the recognition of cancer biomarkers.

Reviewer 2 Report
In this review ''Relevance of miR-223 as Potential Diagnostic and Prognostic Markers in cancer'' authors discuss the general biology of microRNA-223 (miRNA-223), its role in different types of cancer and more specifically in the mechanisms important for cancer progression such as proliferation, apoptosis, and cell cycle regulation, and further emphasize its role as the potential diagnostic and therapeutic biomarker.
Broad comments: This manuscript is inadequately structured and poorly written, with numerous logical and technical errors.
Specific comments:
- The authors have failed to cite the most recent and relevant papers on the miRNA and its role in cancer. Further, many references are wrongly cited which makes this manuscript rather difficult to read and additionally, after some statements authors often do not include references. Some of the examples are as follows: line 25, more recent overview on miRNA should be cited (PMID: 32582699), line 54 a paper on myeloid lineage PMID: 21725054 should be cited, more recent data on hepatocellular carcinoma (eg. PMID: 32330203), breast cancer (eg. PMID: 33928027), lung cancer (eg. PMID: 31488416), cell cycle (eg. PMID: 20826802), cell proliferation and invasion (eg. PMID: 30776580), miRNA as biomarkers (eg. PMID: 31979244), plasma and stool miRNA (eg PMID: 26848774)… No references in line 58, no references in line 60 (eg PMID: 30878527, PMID: 34956210), 62 63… In line 81 authors say CLL and lung cancer but they cite acute lymphoblastic leukemia and gastric cancer papers, line 152 wrong reference (30), lines 157/158 authors say colon cancer but they cite ovarian and laryngeal cancer papers. References are inadequately cited at the end; journal’s names are missing and some years are wrongly cited (eg. Ref 49).
- Authors should in more details discuss the role of miRNA-223 in hematological malignancies (leukemia, lymphoma…); there is almost no mention on that matter
- Authors should in more detail discuss the role of miRNA in cell cycle regulation (cyclins…) and apoptosis
- Authors should provide figures/tables to make this review more readable: eg. one figure on the biogenesis and mechanism of action of miRNA-223 and another figure/table on the role on the miRNA-223 in different cancers as well as role in different signaling pathways.
- The structure of the article in not logical, the authors discuss about different carcinomas and then they say almost the same in the part about therapeutic biomarkers.
Author Response
Reviewer 2.
Comment 1. The authors have failed to cite the most recent and relevant papers on the miRNA and its role in cancer. Further, many references are wrongly cited which makes this manuscript rather difficult to read and additionally, after some statements authors often do not include references. Some of the examples are as follows: line 25, more recent overview on miRNA should be cited (PMID: 32582699), line 54 a paper on myeloid lineage PMID: 21725054 should be cited, more recent data.on hepatocellular carcinoma (eg. PMID: 32330203), breast cancer (eg. PMID: 33928027), lung cancer (eg. PMID: 31488416), cell cycle (eg. PMID: 20826802), cell proliferation and invasion (eg. PMID: 30776580), miRNA as biomarkers (eg. PMID: 31979244), plasma and stool miRNA (eg PMID: 26848774)… No references in line 58, no references in line 60 (eg PMID: 30878527, PMID: 34956210), 62 63… In line 81 authors say CLL and lung cancer but they cite acute lymphoblastic leukemia and gastric cancer papers, line 152 wrong reference (30), lines 157/158 authors say colon cancer but they cite ovarian and laryngeal cancer papers. References are inadequately cited at the end; journal’s names are missing and some years are wrongly cited (eg. Ref 49).
Response 1. Thank you for your questions and comments and concerns. We have added the recent references in their respected section as well as added the detailed data of hepatocellular carcinoma, breast cancer, lung cancer, cell cycle, cell proliferation and invasion, miRNA as biomarkers, plasma and stool miRNA by using reviewers suggested references.
Comment 2. Authors should in more details discuss the role of miRNA-223 in hematological malignancies (leukemia, lymphoma…); there is almost no mention on that matter
Response 2. Thank you for your kind suggestions. We have modified discuss the role of the miRNA-223 in hematological malignancies e.g. leukemia and lymphoma by using the latest references.
“It was reported that miR-223 functions as oncomiR in T-cell acute lymphoblastic leukemia, whereas in acute myeloid leukemia (AML) functions as a tumor suppressors [34a][34b][34c]. According to Gao et al., low expression of miR-223 has been found in various human hematologic malignancies and solid tumors that is acting as tumor-suppressor and has a significant role in leukemia and lymphomas. miR-223 significantly inhibit proliferation, growth rate, colony formation of cells in vitro, and tumor formation in vivo through targeting IGF-1R and its downstream PI3K/Akt/mTOR/p70S6K pathway in leukemia [34b][34d][34e]”.
References:
34a. Fazi F, Racanicchi S, Zardo G, Starnes LM, Mancini M, Travaglini L, Diverio D, Ammatuna E, Cimino G, Lo-Coco F, Grignani F, Nervi C. Epigenetic silencing of the myelopoiesis regulator microRNA-223 by the AML1/ETO oncoprotein. Cancer Cell. 2007, 12, 457-66. doi: 10.1016/j.ccr.2007.09.020.
34b. Gao Y, Lin L, Li T, Yang J, Wei Y. The role of miRNA-223 in cancer: Function, diagnosis and therapy. Gene. 2017, 616, 1-7. doi: 10.1016/j.gene.2017.03.021.
34c. Mavrakis, K., Van Der Meulen, J., Wolfe, A. et al. A cooperative microRNA-tumor suppressor gene network in acute T-cell lymphoblastic leukemia (T-ALL). Nat Genet. 2011, 43, 673–678 (2011).
https://doi.org/10.1038/ng.858.
34d. Zhao, F., Han, J., Chen, X., Wang, J., Wang, X., Sun, J., & Chen, Z. (2016). miR-223 enhances the sensitivity of non-small cell lung cancer cells to erlotinib by targeting the insulin-like growth factor-1 receptor. International Journal of Molecular Medicine. 2016, 38, 183-191. https://doi.org/10.3892/ijmm.2016.2588.
34e. Tachibana H, Sho R, Takeda Y, Zhang X, Yoshida Y, Narimatsu H, Otani K, Ishikawa S, Fukao A, Asao H, Iino M. Circulating miR-223 in Oral Cancer: Its Potential as a Novel Diagnostic Biomarker and Therapeutic Target. PLoS One. 2016, 11, e0159693. doi: 10.1371/journal.pone.0159693.
Comment 3. Authors should in more detail discuss the role of miRNA in cell cycle regulation (cyclins…) and apoptosis.
Response 3. Thank you for your kind suggestions. We have discussed the role of miRNA in cell cycle regulation by using recent references.
“According to Xu et al., overexpression of miR-223 leads to reduced FBXW7 mRNA level and increases endogenous cyclin E protein which in-turn increases genomic instability. In contrast, a low level of miR-223 level leads to increased Fbw7 expression and decreased cyclin E activity. Furthermore, it was proved that miR223 expression is reactive to acute alterations in cyclin E regulation by the Fbe7 pathway [41a]”.
References:
41a. Liu Z, Ma T, Duan J, Liu X, Liu L. MicroRNA‑223‑induced inhibition of the FBXW7 gene affects the proliferation and apoptosis of colorectal cancer cells via the Notch and Akt/mTOR pathways. Mol Med Rep. 2021, 23, 154. doi: 10.3892/mmr.2020.11793.
Comment 4. Authors should provide figures/tables to make this review more readable: eg. one figure on the biogenesis and mechanism of action of miRNA-223 and another figure/table on the role on the miRNA-223 in different cancers as well as role in different signaling pathways.
Response 4. Thank you for your valuable advice. We have made a table summarizing the reviewed knowledge. Furthermore, we have drawn a figure explaining biological pathways by getting inspiration from the suggested recent review.
Comment 5. The structure of the article in not logical, the authors discuss about different carcinomas and then they say almost the same in the part about therapeutic biomarkers.
Response 5. Thank you for your valuable advice. We have modified the sentence structure and added some more literature by using the latest references.

Reviewer 3 Report
The authors provided an overview of the tumor suppression or oncogenic role of miR-223 in cancers. The included information is large. However, the organization of the manuscript could be improved. For example, the authors could divide the sections according to the tumor suppression and oncogenic functions of miR-223 rather than divide according to cancer types. In addition, it suggests the authors summarize a table to present the role of miR-223 in cancers, including the targets, functions, and diagnostic potentials.
Author Response
Reviewer 3.
Overall comment. The authors provided an overview of the tumor suppression or oncogenic role of miR-223 in cancers. The included information is large. However, the organization of the manuscript could be improved. For example, the authors could divide the sections according to the tumor suppression and oncogenic functions of miR-223 rather than divide according to cancer types. In addition, it suggests the authors summarize a table to present the role of miR-223 in cancers, including the targets, functions, and diagnostic potentials.
Responses. Thank you for your kind suggestion. We have modified the literature and specifically mentioned the role of miR-223 as tumor suppressor and oncogene at their respected place. Furthermore, we have made a table summarizing the reviewed knowledge, as well as drawn a figure explaining biological pathways by getting inspiration from the suggested recent review.

Round 2
Reviewer 2 Report
Authors have addressed all the raised issues and no further revisions are needed.